# Results of a fast-track HIV and hepatitis C screening protocol in Barcelona, Spain

Iván Chivite[1,2]*, Vanessa Guilera[1], Pilar Callau[1], Raquel Aguiló[1], Elisa de Lazzari[1,2,3], M.J. Merino[1], Laia Diaz[1], Alba Carrodeguas[4], José Luis González-Sánchez[4], Josep Mallolas[1,2,3]

1 Hospital Clínic de Barcelona, Barcelona, Spain, 2 Instituto de Investigaciones Biomédicas August Pi i Sunyer (IDIBAPS), Spain, 3 CIBER de Enfermedades Infecciosas (CIBERINFEC), Instituto de Salud Carlos III, Madrid, Spain, 4 Gilead Sciences, Spain

* ichivite@clinic.cat

## Abstract

### Introduction

Human immunodeficiency virus (HIV) and hepatitis C virus (HCV) continue to be a significant public health concern. Screening is a critical strategy for HIV and HCV control to reach the World Health Organization's elimination goals by 2030. This study assessed the outcomes of a healthcare quality improvement project integrating routine opportunistic BBV screening and linkage to care in emergency services for high-risk patients. This project aimed at providing HIV and HCV fast-track screening among patients seeking care in the emergency department of Hospital Clínic de Barcelona (Spain) and re-engage individuals previously diagnosed but not currently in care.

### Methods

This observational study included patients ≥18 years old who presented to the emergency department reporting genitourinary complaints or recent high-risk exposures for HIV, HCV, or other sexually transmitted infections. Using the FOCUS TEST model as a framework, a systematic opportunistic HIV and HCV screening and offered linkage to care (LTC) to patients with positive test results was conducted. Screening was performed using fourth-generation chemiluminescence immunoassays for HIV and HCV antibodies, including p24 antigen detection, with confirmatory HCV RNA testing by PCR. The FOCUS TEST model supports automatic integration of testing into routine workflows and staff training. For each of these blood-borne viruses, the screening volume, testing uptake, seroprevalence, characteristics of patients with new infections, and LTC rates were analyzed.

**Data availability statement:** All relevant data are within the paper and its Supporting information files.

**Funding:** We acknowledge funding from Gilead Sciences' FOCUS program to support HIV and viral hepatitis screening and linkage to the first medical appointment after diagnosis. FOCUS funding does not support activities beyond the first medical appointment and is agnostic to how organizations handle subsequent patient care and treatment. The funders had no role in study design, data collection and analysis, decision to publish, or preparation of the manuscript.

**Competing interests:** Alba Carrodeguas and José Luis González-Sánchez own stock in and are employees of Gilead Sciences. The remaining authors declare no conflicts of interest concerning the research, authorship, or publication of this article. Data collection and management were conducted independently, with additional oversight of independent data monitoring agencies. This does not alter our adherence to PLOS ONE policies on sharing data and materials. There are no patents, products in development or marketed products associated with this research to declare.

**Abbreviation:** Ab, Antibody; AIDS, Acquired immunodeficiency syndrome; ECDC, European Centre for Disease Prevention and Control; HBV, Hepatitis B virus; HCV, Hepatitis C virus; HIV, Human immunodeficiency virus; NA, Not available; RNA, Ribonucleic acid; STI, Sexually transmitted infection; WHO, World Health Organization.

## Results

Between January 2020 and December 2022, 35,285 blood-borne virus tests were performed. The number of new infections detected was 38 for HIV (0.41% seroprevalence) and 34 for HCV (0.19% HCV RNA prevalence). LTC was achieved for 89% and 100% of patients diagnosed with new HIV and HCV infections, respectively. A separate set of patients with HIV (n = 297) or HCV (n = 25) infections identified prior to this screening program were re-linked to care via this project.

## Conclusion

This healthcare quality improvement project was feasible and successful in achieving its goal of providing systematic opportunistic HIV and HCV screening to patients seeking urgent care. Importantly, the program also enabled LTC of a considerable number of patients previously diagnosed but not retained in care, further strengthening its impact on public health. These outcomes align with global goals for the elimination of HIV and HCV as public health threats by 2030 and demonstrate that similar fast-track screening and linkage strategies could be effectively implemented in other urban emergency settings with appropriate infrastructure and support.

## Introduction

At the end of 2022, the World Health Organization (WHO) reported that an estimated 39.0 million people globally were living with human immunodeficiency virus (HIV) and 1.3 million people had become newly infected in the same year [1]. In Europe, over 110,000 HIV diagnoses were made in 2022 [2]. Similarly, hepatitis C virus (HCV) continues to be a major public health concern, with 13,914 HCV infections recorded in 2020 in Europe alone [3].

In Spain, a 2017–2018 study by the Ministry of Health estimated the seroprevalence of HIV and HCV infections in the general population at 0.13% for HIV (of which 13% corresponded to newly diagnosed infections) and 0.22% for active HCV infection [4]. A more recent HIV survey by the Ministry of Health in 2021 found an estimated 0.31% HIV prevalence in the Spanish general population [5]. More specifically, in the region of Catalonia, approximately 33,700 people were living with HIV in 2019 (0.43% prevalence among the adult population), and 329 new HIV diagnoses were reported in 2020 [6]. Unfortunately, data for viral hepatitis in Catalonia are scarce and restricted to special populations only [7].

Screening is a critical component in any strategy for HIV and HCV control and elimination of HIV, HCV, and other sexually transmitted infections (STIs) [8]. Not only does it aid in diagnosing these infections and providing patients with access to treatment, but it is also a main pillar of secondary prevention; screening contributes to reducing the prevalence of existing disease at its earliest stage, which helps prevent onward transmission [8].

In hospital settings, the European Centre for Disease Prevention and Control (ECDC) recommends that integrated testing for HIV/HCV be considered in patients who are diagnosed with any one of the three infections, belong to certain risk groups, have STI-compatible signs or symptoms, or who present with an HIV indicator condition [8]. In areas of high prevalence or incidence of STIs (and those with intermediate HCV prevalence or incidence), testing should be considered for any patient attending the emergency department (ED) or being admitted to hospital who undergoes a blood test for another indication and has never been previously tested for HIV/HCV infection [8]. Despite these recommendations, screening in ED settings often remains ad hoc, dependent on individual clinical judgment, and inconsistently implemented. Common limitations include time constraints, lack of staff training, non-integrated workflows, and absence of standardized referral pathways for positive cases. Our fast-track protocol was specifically designed to overcome these barriers by embedding screening within routine workflows, automating eligibility and test ordering through electronic systems, and assigning dedicated staff to facilitate immediate linkage to care (LTC).

EDs are particularly important in the context of STI screening. With their unrestricted access, EDs frequently act as safety nets for patients with HIV and viral hepatitis or those at high risk for these infections, who may lack optimal links with primary care providers due to health inequities or other reasons. Evidence shows that blood-borne virus (BBV) screening in ED settings is a feasible and acceptable option to identify new HIV and viral hepatitis infections [9]. Opt-out testing, automated laboratory order algorithms, and electronic alerts are associated with a higher testing uptake [10–14]. Although LTC upon BBV screening in EDs is still sub-optimal, screening in this setting could be used to re-engage patients who are already aware of their infection but not yet linked to care [14–16]. Challenges to effective LTC include lack of patient awareness of the importance of follow-up, structural barriers such as transportation or clinic accessibility, stigma associated with infection status, and insufficient coordination between emergency and specialist services. Addressing these obstacles requires dedicated personnel and system-level adaptations, which were incorporated into our study design.

Although the importance of HIV and HCV screening in EDs is well recognized, few studies have implemented integrated strategies combining systematic screening, automation, coinfection assessment, and re-linkage to care. For example, the systematic review by Simmons et al. [9] reported a wide range of HIV and HCV seroprevalence rates in ED settings (up to 13% for HIV in certain contexts), as well as significant variability in acceptability and implementation. One of the key findings was that, although screening was feasible, LTC rates were often suboptimal and inconsistently reported across centers.

Other studies, such as Hoxhaj et al. [13], evaluated opt-out HIV screening using non-rapid technology in high-volume EDs but did not include HCV or report LTC outcomes. Freeman et al. [12] documented good patient acceptance of rapid HIV testing in a southeastern ED but did not address automation or continuity of care. More recently, Vaz-Pinto et al. [14] demonstrated that automated algorithms integrated into the electronic medical record improved early HIV diagnosis in Portuguese EDs; however, their study focused solely on HIV, did not include HCV, and did not report data on re-linkage of previously diagnosed patients. In this context, there remains a clear need for studies that go beyond assessing feasibility and uptake of HIV and HCV screening, and that report comprehensive outcomes including seroprevalence, coinfection rates, LTC, and effective strategies for re-engaging patients lost to follow-up.

Given the global goal of eliminating AIDS, HCV, and other STIs by 2030—and the substantial proportion of individuals who remain undiagnosed or not linked to care—there is a clear need for people-centered, fast-track screening and linkage-to-care strategies [17]. The aim of this study was to assess the results and impact of a healthcare quality improvement project toward providing HIV and HCV screening among patients seeking care in the ED of Hospital Clínic de Barcelona, in Spain, which is the primary healthcare provider for approximately 540,000 residents across several districts of the city, and a referral center for various medical specialties for patients from the broader Catalonia regions of Vallès Oriental and Osona. This context underscores the relevance and reach of implementing systematic screening interventions in such an urban tertiary care setting. Specifically, our objectives were: (1) to determine the uptake of HIV and HCV screening

among eligible patients; (2) to estimate the prevalence of newly diagnosed infections; (3) to describe the demographic and clinical characteristics of patients with new infections; and (4) to provide LTC for both newly diagnosed and previously known but not retained patients.

## Materials and methods

### Health area covered by the study

In January 2020, a project was implemented to provide fast-track screening and immediate referral to specialist care in a novel STI outpatient clinic for adults in Hospital Clínic de Barcelona, a tertiary university hospital serving a large urban population. The screening program was implemented in its emergency department and associated outpatient STI clinic. This hospital is the main healthcare provider for a population of 540,000 inhabitants in the Barcelona districts of Sants-Montjuïc, Les Corts, Sarrià, Sarrià-Sant Gervasi, and Eixample, and is a reference center across a wide range of medical specialties for patients from the Catalonia areas of Vallès Oriental and Osona.

### Patient eligibility

Patients eligible for this project were adults ≥18 years old who presented to the ED of Hospital Clínic de Barcelona with either genitourinary complaints or reporting recent high-risk exposures, such as engaging in chemsex or unprotected penetrative intercourse, or sharing injection materials. Interventions in this project ranged from immediate HIV and HCV blood tests and/or bacterial STI testing to post-exposure prophylaxis and enrolment in comprehensive prevention services (e.g., pre-exposure prophylaxis and harm reduction). Hepatitis B virus (HBV) was not evaluated because data on this infection was not consistently available. However, vaccination status was requested during the first visit, and those who required HBV vaccination were referred to their respective health centers.

### Study design

A retrospective cohort study analyzing the results of this systematic opportunistic HIV and HCV screening program was conducted between 1st January 2020 and 31st December 2022. After testing, appropriate patient follow-up or discharge was performed depending on test results. LTC, defined as confirmation of attendance at a first medical appointment after a positive test result, was managed by dedicated staff, who received real-time notice of positive test results and contacted these patients regarding successful linkage to specialist medical care.

### Ethical considerations

All data analyzed were collected as part of routine medical care, without additional patient contact, and were subsequently anonymized before analysis. Therefore, no identifiable information was used, ensuring compliance with the General Data Protection Regulation (GDPR, EU 2016/679) and the Spanish Organic Law 3/2018 on Personal Data Protection and Digital Rights Guarantee. Access to the data for research purposes was granted on January 15, 2023. Throughout the entire analysis process, the authors did not have access to information that could identify individual participants at any time, either during or after data collection. The data analyzed were fully anonymized prior to being provided to the research group.

According to Article 14 of Spanish Organic Law 3/2018, research using anonymized data for the purposes of public health does not require explicit consent or ethics committee review when there is a justified general interest, as is the case for our study. The implemented screening program in the emergency department aligns with public health strategies for the early detection of bloodborne infections in high-risk populations, in accordance with the WHO recommendations and global objectives for eliminating HIV and HCV as public health concerns by 2030.

It is important to emphasize that our study is retrospective and observational, not experimental, and that the data used do not compromise the integrity, well-being, or autonomy of patients, in line with the principles of the Declaration of

Helsinki. Furthermore, in our institutional and legal context, studies that analyze anonymized secondary data for a public interest purpose and healthcare quality improvement do not require formal ethics committee approval. In conclusion, the study fully complies with ethical and legal requirements, preserving participant confidentiality and ensuring responsible data use within the framework of an intervention that has demonstrated clear benefits in the detection and linkage to care of previously undiagnosed patients.

## Description of the FOCUS TEST model

The FOCUS TEST model to encourage systems change and to extend the adoption of screening and LTC [18] was applied.

"TEST" stands for four pillars: (1) Testing: Screening was embedded into routine ED workflows for eligible patients, using existing clinical staff and sample collection protocols. (2) Electronic health record (EHR) enhancements: The EHR system was modified to include automated prompts and predefined order sets that facilitated rapid test ordering and eligibility checks. (3) Systemic policy change: An institutional protocol was adopted mandating HIV and HCV screening for all patients meeting risk criteria, ensuring consistency across shifts and personnel. (4) Training and feedback: Regular training sessions were conducted for ED clinicians and nurses, accompanied by monthly feedback reports on screening volume, positivity rates, and LTC performance, which helped maintain program engagement and quality. [18].

## Immunoassay tests for screening

Fourth-generation chemiluminescence assays were performed to determine antibodies against HIV and HCV, as well as p24 antigen. Specifically, HIV Ag/Ab Combo (Siemens, Sistema Atellica) and HCV (Siemens, Sistema Atellica). HCV viral load determination was calculated through real-time PCR on the automated system COBAS® AmpliPrep/COBAS® TaqMan® HCV, v2.0 (Roche, Sistema COBAS).

## Variables and outcomes

Variables considered in this study were: age (categorized in five strata: 18–22, 23–30, 31–40, 41–50, 51–90), sex, and injected drug use.

For HIV and HCV, the main outcomes reported in this analysis were: number of tests performed, testing uptake (defined as the percentage of patients undergoing testing out of the total number of patients who were offered it), number of new infections (i.e., tests with a positive result), seroprevalence (defined as the percentage of patients with a positive test result out of the total number of patients who underwent testing), and rate of LTC (defined as the percentage of patients attending a first medical appointment after diagnosis out of the total number of patients with a positive test result).

Other variables and outcomes of interest in this study included: presence of acute infection (for new HIV infections only; defined as having a positive p24 antigen test result); re-LTC (for patients with HIV or HCV infections that were previously detected outside this screening program); and sexual behavior (for patients living with HIV who were re-linked to care only).

These variables were selected based on their clinical and public health relevance and their alignment with international screening and LTC indicators. Demographic factors such as age and sex are known to influence infection risk and care engagement. Uptake, prevalence, and LTC are critical for assessing the effectiveness and operational feasibility of a screening program, and are key metrics in evaluating progress toward WHO elimination targets.

## Statistical analysis

All data points were analyzed descriptively; the resulting statistics are reported as absolute values and/or percentages. To quantify the precision of the testing uptake and seroprevalence rates for HIV Ab, HCV Ab, and HCV RNA screenings, confidence intervals (CIs) were calculated for each of these measures. For comparative analysis, demographic groups were

compared by Pearson's Chi-squared test. P-values were estimated using Monte Carlo simulation with 10,000 replicates to account for small expected counts and ensure statistical robustness. All statistical analyses were performed using R version 4.3.3 (2024-02-29 ucrt).

Data extraction was performed using predefined queries applied to the hospital's electronic health records system. Validation checks were conducted to ensure consistency across variables, and records were cross-referenced with laboratory and appointment databases to confirm test results and linkage to care status. Data cleaning included removal of duplicates, logical consistency verification (e.g., matching test results with timestamps and visit types), and manual review of records with missing or implausible values. Cases with missing data for key variables were excluded from the corresponding sub-analyses.

## Results

### Testing uptake

A combined 35,285 BBV tests in the STI outpatient clinic of Hospital Clínic de Barcelona within the study period were performed. Testing uptake among eligible patients seeking care in this period was 70% (*n* = 9,304/13,201) for HIV Ab and 92% (*n* = 17,313/18,819) for HCV Ab (**Table 1**). Out of all the patients with a positive HCV Ab test result, 66% (*n* = 160/241) underwent HCV RNA testing.

### New HIV infections

Out of the 9,304 patients tested for HIV Ab, 38 new HIV infections were diagnosed, 0.41% (0.28–0.54) seroprevalence (**Table 1**), 17 of which (45%) had a positive p24 antigen test result suggestive of acute or early-phase infection. Most patients diagnosed (79%) were aged 23–40 years; 95% (*n* = 36) were cisgender men; and none reported injection drug use (**Table 2**).

### New HCV infections

Out of the 17,313 patients tested for HCV, 241 patients were identified with an HCV Ab positive result, 1.39% (1.22–1.57) HCV Ab prevalence, among whom 34 were diagnosed with new HCV infections (21.25% RNA positivity rate; 0.19% HCV RNA prevalence) (**Table 1**). Most patients with new HCV infections (85%, *n* = 29) were 23–50 years old and all were cisgender men (**Table 2**). Eighty-five percent (*n* = 29) of the patients with new HCV infections were coinfected with HIV, and 7% (*n* = 2) with HBV. Among patients with an HCV Ab positive result, 11 patients reported injected drug use, and accounted for 7.9% of the patients with available data (**Table 2**).

**Table 1.  Screening volume, testing uptake, and prevalence of HIV and HCV infections in this study.**

| | HIV Ab screening | HCV Ab screening | HCV RNA screening |
|---|---|---|---|
| **Patients offered testing, n** | 13,201 | 18,819 | 241 |
| **Patients undergoing testing, n** | 9,304 | 17,313 | 160 |
| **Testing uptake, % (95% CI)** | 70.48% (69.70-71.26) | 91.99% (91.61-92.39) | 66.39% (60.43-72.35) |
| **Patients with positive results, n** | 38 | 241 | 34 |
| **Seroprevalence and RNA positivity rate, % (95% CI)** | 0.41% (0.28-0.54) | 1.39% (1.22-1.57) | 21.25% (14.91-27.59) |

Testing uptake was calculated as the percentage of patients undergoing testing out of the total number of patients who were offered it.

Seroprevalence was calculated as the percentage of patients with a positive test result out of the total number of patients who underwent testing.

CI: confidence interval; HIV Ab: HIV antibody; HCV Ab: hepatitis C virus antibody; HCV RNA: hepatitis C virus ribonucleic acid.

**Table 2. Characteristics of patients with new HIV and HCV infections.**

| Patient characteristic, n (%) | New HIV infections identified in this screening N=38 | HIV infections previously identified outside this screening N=298 | HCV (Ab+) results identified in this screening N=241 | New HCV infections (RNA+) identified in this screening N=34 | Pearson's Chi-squared test* | p-value* |
|---|---|---|---|---|---|---|
| **Age in years** | | | | | 79.031 | <0.001 |
| 18–22 | 2 (5%) | 10 (3%) | 1 (0%) | 1 (3%) | | |
| 23–30 | 11 (29%) | 107 (36%) | 27 (11%) | 7 (21%) | | |
| 31–40 | 19 (50%) | 111 (37%) | 90 (37%) | 12 (35%) | | |
| 41–50 | 4 (11%) | 43 (14%) | 63 (26%) | 10 (29%) | | |
| 51–90 | 2 (5%) | 27 (9%) | 60 (25%) | 4 (12%) | | |
| **Gender identity** | | | | | 31.262 | <0.001 |
| Cisgender man | 36 (95%) | 238 (80%) | 222 (92%) | 34 (100%) | | |
| Cisgender woman | 1 (3%) | 18 (6%) | 12 (5%) | 0 (0%) | | |
| Transgender person[a] | 1 (3%) | 42 (14%) | 7 (3%) | 0 (0%) | | |
| **Injected drug use** | | | | | 103.26 | <0.001 |
| Yes | 0 (0%) | 5 (2%) | 11 (7%) | NA | | |
| No | 35 (92%) | 269 (90%) | 129 (83%) | NA | | |
| Unknown | 3 (8%) | 24 (8%) | 101 (10%) | NA | | |

[a]Gender information not available. HIV Ab: HIV antibody; HCV Ab: hepatitis C virus antibody; HCV RNA: hepatitis C virus ribonucleic acid; NA, not available.

*Pearson's Chi-squared test, p-value was estimated by Monte Carlo simulation (10000 replicates)

### LTC and retrieval of patients lost to follow-up

LTC was achieved for 89% (n = 34) and 100% (n = 34) of patients diagnosed in this study with new HIV and HCV infections, respectively.

Besides the patients who were tested as part of this study, individuals with known HIV or HCV infections who had not been engaged in care at the time of their visit were identified. These individuals were detected through routine serological testing or existing electronic health record alerts and were re-engaged via the same linkage mechanisms used in the screening protocol. A total of 298 HIV Ab-positive and 25 HCV Ab-positive patients were successfully re-linked to specialist care. The characteristics of the patients with previously identified HIV infection who were lost to follow-up and re-linked to care are shown in **Table 2**. Of this group, 21% (n = 54) reported heterosexual practices, 79% (n = 204) were men who have sex with men, and 2.5% (n = 5) reported injected drug use.

## Discussion

The healthcare quality improvement project described herein was successful in attaining its main goal of providing HIV and HCV screening among patients seeking urgent care in the ED of Hospital Clínic de Barcelona, demonstrating the feasibility of the approach.

The prevalence rates of previously undiagnosed HIV and HCV infections in this study were very similar compared with those estimated for the general population in Spain: 0.41% vs 0.31% for HIV infection and 0.19% vs 0.22% for HCV infection [4,5]. The slightly higher HIV rate is not surprising considering that patients eligible for this study were attending an ED and presented with genitourinary complaints or reported recent high-risk behaviors associated with these infections. In fact, seroprevalence rates reported in the literature among populations in different countries who attend an ED are also notably high, falling within the range 0.8%–13% for HIV and 1.5%–17% for HCV infections (the latter including antibody

and RNA prevalence studies) [9]. For HIV in particular, the 0.41% seroprevalence rate found in this study is very close to the 0.43% prevalence rate estimated for the region of Catalonia (where Barcelona is located) in 2019 [6].

Compared to other programs implemented in emergency departments (EDs), our detection rates fall within the low to intermediate range reported internationally. For instance, Simmons et al. conducted a systematic review on blood-borne virus screening in EDs, finding HIV prevalence rates ranging from 0.8% to 13%, and up to 17% for HCV [9]. Most European studies included in that review reported rates closer to 1%, making our findings consistent with an urban setting with a moderate infection burden. In Portugal, Vaz-Pinto and colleagues implemented an automated HIV screening program in EDs and found a 0.55% prevalence, which is very similar to ours [14].

Most patients with new HIV infections in this study were men aged 23–40 years. Prevalence data among the general population in Spain in 2017–2018 showed that most HIV-positive cases corresponded to men aged 35 years and older [4]. Data collected in 2021 in Barcelona, where Hospital Clínic de Barcelona is located, also coincide in that the majority of patients with HIV infection were men, but show that 67.5% of the patients newly diagnosed were younger than 40 years [19]. Therefore, the most prevalent age categories for HIV infections in our study are consistent with recent data from Barcelona, but slightly younger than those in the general population in Spain in 2017–2018 [4,19].

All patients with active HCV infection in this study were men, which is in line with the remarkably higher HCV prevalence in men versus women that has already been reported in the general population in Spain (0.35% versus 0.08%) [4]. Regarding age groups, while the highest HCV prevalence among the general population was found in those aged 50–59 years old, this study detected the greatest HCV prevalence among patients aged 23–59 years [4]. Again, the different populations included in either study could account for this slight discrepancy in age categories.

Injected drug use, a known high-risk exposure behavior for HIV and HCV [8,17,20,21], was reported in 7% of patients with HCV infection in this study. However, none of the patients with new HIV infection reported using injected drugs. Data from Barcelona revealed injected drug use as the mode of transmission in only 2.1% ($n = 3$) of all new HIV infections in 2021 among men ($N = 146$) and 0% ($n = 0$) among women ($N = 23$) [19]. With such a low percentage of injected drug use reported in published data from Barcelona, it is not surprising that no cases of injected drug use among HIV diagnoses were detected, as the overall number of new HIV infections was much smaller in this study ($N = 34$).

Beyond detecting new infections, a key strength of this program was the ability to re-engage patients with previously known HIV or HCV diagnoses who had disengaged from care. This aligns with global health strategies that emphasize re-linkage as an essential element of the continuum of care, especially in high-throughput settings like EDs where such patients may sporadically access the healthcare system. According to ECDC guidelines, all testing programs for HIV and HCV must include a referral pathway to link diagnosed patients to appropriate treatment and care services; however, LTC is recognized as a particular challenge [8]. Patients with HIV or HCV include people with different levels of familiarity with care services and access to them [22–25]. The inclusion of dedicated personnel to facilitate care navigation likely contributed to the high LTC rates observed—89% for HIV and 100% for HCV—which fall at the upper end of the 21%–100% range reported in ED-based BBV testing studies [9,14–16]. In addition, HIV and HCV screening in ED settings could also help identify and re-engage patients who are aware of their infection but not yet linked to care. In this study, re-LTC was successfully achieved for 297 patients who were HIV Ab positive and 25 who were HCV Ab positive as previously diagnosed outside this study.

In the context of care services for patients with HIV and HCV, numerous challenges remain to be overcome, for which several strategic actions have been recommended with the aim of improving the continuum of care and ultimately ending these epidemics [17,26]. In particular, as patients proceed along the healthcare pathway of prevention, diagnosis, care and treatment services for HIV and HCV, a number of patients may abandon or lose access to services at each step of the follow-up process. The WHO global health sector strategies on HIV and viral hepatitis stress the importance of organizing and expanding services in a people-centered manner, such that individuals can receive appropriate HIV testing, have early access to and engagement in care, can be re-engaged if lost at some point in the continuum of care, and receive

treatment and continuous care throughout different stages in life [17]. In this regard, HIV and other BBV screening programs in ED settings, such as the one described in this study, offer a unique possibility to reach populations 1) who may not consider themselves at risk; 2) who may face barriers that prevent them from seeking testing and care; and/or 3) who may have been diagnosed but were never linked to care or lost engagement in care [9,14]. The results of our fast-track screening and referral program indeed show high linkage rates for new infections and considerable numbers of previously diagnosed patients being re-engaged with care services, which represent positive outcomes in light of the current recommendations and goals aimed at ending HIV and viral hepatitis epidemics. These findings support the role of EDs and hospital-based STI clinics as critical touchpoints for identifying individuals who may not be reached through conventional screening in primary care settings. Our model also demonstrates that integrating LTC mechanisms into routine hospital workflows can achieve performance levels that meet or exceed global public health benchmarks.

In terms of broader implications, our study highlights that systematized, opportunistic BBV screening in urban ED settings is feasible and highly effective in identifying both new and previously known cases requiring care. This approach can serve as a replicable model for other healthcare systems, particularly in urban centers, willing to operationalize the WHO 2030 targets for HIV and HCV elimination. From a clinical standpoint, embedding these protocols helps reduce missed diagnostic opportunities and facilitates earlier engagement in care, ultimately improving long-term patient outcomes and reducing community transmission.

This study has some limitations that should be considered to appropriately interpret the findings. First, data were collected at a single site in Barcelona, which may limit the generalizability of our findings to other regions with different sociodemographic characteristics or different epidemiology of HIV and HCV infections. This may be particularly relevant in rural or non-European contexts where emergency department utilization patterns and access to STI/HIV services differ significantly. Second, due to the study setting and eligibility criteria, the population analyzed consisted of patients presenting with high-risk factors or exposed to high-risk behaviors. This could have introduced selection bias to some extent, and therefore our findings may not necessarily be representative of the general population (as previously discussed). Additionally, individuals who declined testing may differ systematically from those who accepted, potentially affecting the observed seroprevalence and LTC rates. Third, no exhaustive data on specific risk factors for HIV and HCV transmission in the study population were collected, which may limit our ability to tailor prevention and treatment interventions based on individual risk profiles. Finally, it was not within the scope of this project to collect data on the long-term outcomes of patients diagnosed with HIV and HCV, such as virologic suppression and liver fibrosis staging.

## Conclusion

Our study shows positive outcomes for a fast-track screening and LTC protocol for HIV and viral hepatitis. Based on our findings, we recommend that healthcare institutions implement routine, opt-out HIV and HCV screening in emergency departments using automated testing algorithms and electronic medical record alerts. Additionally, the inclusion of dedicated staff responsible for real-time follow-up and LTC coordination is essential. Policymakers should prioritize funding and resource allocation to support such programs, which are critical for achieving national and global viral hepatitis and HIV elimination goals. The high LTC rates achieved in this study indicate that immediate referral to specialist care is feasible in this setting using dedicated staff. LTC is critical in ensuring that newly diagnosed patients and those who were previously diagnosed but lost engagement in care receive appropriate treatment and follow-up. Future research should focus on evaluating the long-term clinical outcomes of patients diagnosed through fast-track screening, including retention in care and treatment success rates. Additionally, studies exploring cost-effectiveness, patient acceptability, and the implementation of similar programs in diverse healthcare settings would further inform the scalability and sustainability of such interventions. Investigating barriers to screening uptake among subpopulations less likely to engage in testing could also enhance targeted strategies.

## Author contributions

**Conceptualization:** Iván Chivite, Alba Carrodeguas, José Luis González-Sánchez.

**Data curation:** M.J. Merino, Laia Díaz.

**Formal analysis:** Vanessa Guilera, Pilar Callau, Raquel Aguiló, Elisa de Lazzari, M.J. Merino, Laia Díaz.

**Investigation:** Vanessa Guilera, Pilar Callau, Elisa de Lazzari.

**Methodology:** Raquel Aguiló.

**Resources:** Iván Chivite, Josep Mallolas.

**Supervision:** Iván Chivite, Josep Mallolas.

**Writing – original draft:** Iván Chivite.

**Writing – review & editing:** Iván Chivite, Vanessa Guilera, Pilar Callau, Raquel Aguiló, Elisa de Lazzari, Alba Carrodeguas, José Luis González-Sánchez, Josep Mallolas.

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
