## [Decision Letter · Decision Letter 0]

13 May 2025

PONE-D-24-57558Results of a fast-track HIV and hepatitis C screening protocol in Barcelona, SpainPLOS ONE

Dear Dr. Chivite,

Thank you for submitting your manuscript to PLOS ONE. After careful consideration, we feel that it has merit but does not fully meet PLOS ONE’s publication criteria as it currently stands. Therefore, we invite you to submit a revised version of the manuscript that addresses the points raised during the review process.

Your manuscript was reviewed by one expert in the field. The reviewer identified many important problems in your submission. Please carefully consider the attached comments and provide point-by-point responses.  

We look forward to receiving your revised manuscript.

Kind regards,

Yury E Khudyakov, PhD

Academic Editor

PLOS ONE

Journal Requirements:

2. You indicated that ethical approval was not necessary for your study. We understand that the framework for ethical oversight requirements for studies of this type may differ depending on the setting and we would appreciate some further clarification regarding your research. Could you please provide further details on why your study is exempt from the need for approval and confirmation from your institutional review board or research ethics committee (e.g., in the form of a letter or email correspondence) that ethics review was not necessary for this study? Please include a copy of the correspondence as an "Other" file.

[Gilead FOCUS Program]. 

[Alba Carrodeguas and José Luis González-Sánchez own stock in and are employees of Gilead Sciences. The remaining authors declare no conflicts of interest concerning the research, authorship, or publication of this article. Data collection and management were conducted independently, with additional oversight of independent data monitoring agencies.].

5. We note that your Data Availability Statement is currently as follows:

[All relevant data are within the manuscript and its Supporting Information files.]

6. Please amend the manuscript submission data (via Edit Submission) to include author Iván Chivite Ferriz.

7. Please amend your authorship list in your manuscript file to include author Iván Chivite.

Reviewers' comments:

Reviewer's Responses to Questions

**Comments to the Author**

1. Is the manuscript technically sound, and do the data support the conclusions?

Reviewer #1: Yes

2. Has the statistical analysis been performed appropriately and rigorously? 

Reviewer #1: Yes

3. Have the authors made all data underlying the findings in their manuscript fully available?

Reviewer #1: No

4. Is the manuscript presented in an intelligible fashion and written in standard English?

Reviewer #1: Yes

5. Review Comments to the Author

Reviewer #1: Results of a fast-track HIV and hepatitis C screening protocol in Barcelona, Spain

1. Summary of the Research

The study assessed the outcomes of a fast-track HIV and hepatitis C (HCV) screening protocol implemented in the emergency department of Hospital Clínic de Barcelona. The authors report that they conducted a retrospective observational study, with subjects being adults presenting with genitourinary complaints or high-risk exposures. They used the FOCUS TEST model and achieved a testing uptake of 70% for HIV and 92% for HCV, identifying 38 new HIV infections and 34 new HCV infections. Linkage to care was successfully achieved for 89% of new HIV cases and 100% of new HCV cases. The study highlights the feasibility and effectiveness of systematic opportunistic screening in emergency settings, emphasizing the need for dedicated personnel to ensure high linkage to care rates.

The manuscript has a number of issues that need to be addressed. I recommend that the manuscript needs major revisions.

2. Examples and Evidence

2.1. Major Issues

2.1.1. The Abstract

• Lack of Specificity on the model: The abstract mentions use of the FOCUS TEST model but does not provide information on the model. It is suggested that the authors include a very brief description of the FOCUS TEST model to help readers understand the framework and its application.

• No description of the project in the abstract: The abstract refers to the project but does not clearly indicate what the project is. The authors could include a very short description of the project in the abstract to help readers understand the context and objectives of the study more clearly.

• Patients re-linked to care are not adequately addressed in the abstract: The separate set of patients re-linked to care are only mentioned in the results section of the abstract, which limits the reader's understanding of the full scope and impact of the study. It is proposed that the authors briefly mention this group in the abstract introduction to set the context, in the abstract methods section to explain the process of re-linking these patients to care, and in the abstract conclusion to highlight re-engaging previously diagnosed patients.

• Conclusion too brief and non-specific: The conclusion does not explicitly connect the project's success to broader public health goals and its potential for scalability or application in other settings is not addressed. It is suggested that the authors link the findings to HIV and HCV elimination goals and include the potential for broader application of the findings.

2.1.2. Introduction

• Specificity of Previous Studies: General statistics and the importance of screening are mentioned, but the study does not describe in more detail specific previous studies that have addressed similar issues. Highlighting specific gaps or limitations in past research could better justify the need for this study.

• Detailed Discussion on Linkage to care (LTC) Challenges: Although the introduction mentions that LTC remains sub-optimal, it could benefit from a more detailed discussion on why LTC is challenging and what specific barriers exist, which would provide a clearer context for the study's focus on improving LTC rates.

• Lack of Specificity in Objectives: The introduction mentions the aim of the project but does not specify the specific objectives or hypotheses. The authors could more clearly define the specific objectives or hypotheses of the study.

• Limited Context on Screening Protocols: The importance of screening is briefly mentioned, but the context on existing screening protocols and their limitations is not indicated. The authors could provide more background information on current HIV and HCV screening protocols, their limitations, and how the fast-track protocol addresses these issues, which would help readers better understand the significance of the study.

2.1.3. Materials and Methods

• Study Design: The authors call this a retrospective observational study without specifying the specific type of observational study. This study would however fit better the definition of a retrospective cohort study because it involves identifying a cohort of patients based on their exposure status and then looking back at their medical records to assess outcomes related to HIV and HCV screening and care. I would suggest that the authors consider revising the type of study design accordingly.

• Data Collection and Management: The methods for data collection and management are described, but there is no mention of how data quality and integrity were ensured. The authors could add more information on the procedures for ensuring data quality and integrity, such as validation checks, data cleaning processes, and handling missing data.

• Health Area covered by the study: Most of the information in this sub-section may not be appropriate to be placed in the section. It is suggested that the authors move the detailed description of the health area covered by the study to the background section to help readers understand the context and relevance of the study location. The authors can however retain the concise description of the area of study, sites, and source population that is directly related to the study methodology in the methods section.

• Statistical Analysis: The statistical analysis section is very brief and lacks detail on the specific statistical methods used. The authors should consider including details of the specific statistical tests and methods used, e.g. descriptive statistics (measures of central tendency and dispersion, and frequency distribution for categorical variables), chi-square tests, logistic regression or survival analysis. The authors could also specify the software and tools employed, e.g. SPSS (with version), R (with version) etc.

2.1.4. Results

• Lack of Statistical Detail: The results section lacks detailed statistical analysis, including p-values, other specific statistical tests, etc. I would recommend that the authors include detailed statistical results for each comparison, such as p-values, and specify the statistical tests used (e.g., chi-square test, t-test, etc).

• Comparative Analysis: Comparative analysis is not provided between different groups (e.g., age groups, gender) for new infections and LTC rates. The authors should consider these comparative analyses between different demographic groups and provide statistical significance for these comparisons.

• Definition of acute HIV infection: The authors used positive p24 antigen to determine acute HIV infection. However, as the usual way to confirm acute HIV infection with p24 antigen is together with a negative or indeterminate HIV antibody test result, the authors should consider revising the manuscript accordingly.

• Patients re-linked to care: There is no context provided for the patients who were tested outside the screening program. It is also not clear why the authors reported on the indicators used. There is need for the authors to revise this section, providing context and presenting indicators that align better with the objectives of the study.

2.1.5. Discussion

• Lack of Detailed Comparative Analysis: Though the authors compared the findings with some previous studies, this is inadequate, and in most cases, little attempt to identify similarities and differences together with reasons for any differences. It is recommended that the authors address these issues.

• Context and interpretation of findings: There is inadequate context or interpretation of the findings. The authors could provide more context for the findings, explaining their relevance to public health and clinical practice and discussing the broader implications of the study results.

2.1.6. Conclusion

• Lack of Specific Recommendations: The conclusion does not provide specific recommendations for practice or policy based on the study findings. The authors could include clear and actionable recommendations for healthcare providers and policymakers to improve HIV and HCV screening and linkage to care (LTC) in emergency department settings.

• Absence of Future Research Directions: The directions for future research are not mentioned. The authors could highlight specific areas for future research.

2.1.7. General comments: Language and Structure

• Though the manuscript is written predominantly in the third person, there are some areas where the first person is used. It is recommended that the authors revise the manuscript to ensure that it is consistently written in the third person.

• The terminology used in the manuscript is not always consistent. The authors should review the manuscript and ensure consistency in the terminology throughout the document.

• The manuscript generally does not flow well, and it has a number of grammatical errors. Some sentences are also long and winding, and thus unclear, for example lines 68-72, lines 91—95 and paragraph with lines 199-206. The authors will need to proofread all sections of the manuscript and revise it to improve the language, structure, clarity and general flow.

2.2. Minor Issues

2.2.1. Abstract

• HIV and HCV tests used were not detailed: The abstract mentions that HIV and HCV screening were conducted but does not specify the tests used. It is recommended that the authors mention the tests carried out to provide readers with a clearer understanding of the study methodology.

2.2.2. Introduction

• Insufficient Justification for Study Setting: The choice of Hospital Clínic de Barcelona as the study setting is mentioned but not justified. It is suggested that the authors explain why this particular hospital was chosen for the study, the patient demographics, previous screening efforts and relevance to the study objectives.

2.2.3. Materials and Methods

• The FOCUS TEST Model: The description of the FOCUS TEST model is informative but does not show how the pillars specifically contributed to the study. The authors could expand the description to include examples of how each pillar was implemented in the study.

• Immunoassay Tests for Screening: The authors mention the immunoassays that they used but did not provide specific details on these immunoassays. As it is generally preferred in published literature to provide specific details about the immunoassay tests used as this enhances transparency, reproducibility, and allows for better comparison across studies, it is suggested that the authors provide the names of the assays and their manufacturers.

• Variables and Outcomes: The variables and outcomes are listed, but the rationale for selecting these specific variables is not discussed. The authors could provide a justification for the selection of these variables, explaining their relevance to the study objectives.

2.2.4. Results

• Presentation of Data: The presentation of data in the text is not well structured and is not easy to follow. It is recommended that the authors improve description and presentation of the data in the text.

• Presentation of data in Table 2: The text description of the data in Table 2 is not clear, and the authors could revise the text to ensure a more complete description of the results and improved flow.

2.2.5. Discussion

• Clarity and Structure: The discussion section is not well structured and does not flow well. It is recommended that the authors revise the discussion section to ensure that it flows better and improves clarity.

• Insufficient Explanation of Limitations: Though the limitations of the study are enumerated, they are not discussed in sufficient detail. The authors could expand on the limitations, providing a more comprehensive explanation of how they might affect the study results and interpretations. Potential biases and confounding factors could also be discussed in greater detail.

2.2.6. Conclusion

• Insufficient Summary of Key Findings: The conclusion does not adequately summarize the key study findings. The authors could provide a concise summary of the main results.

6. PLOS authors have the option to publish the peer review history of their article (what does this mean? ). If published, this will include your full peer review and any attached files.

**Do you want your identity to be public for this peer review?** For information about this choice, including consent withdrawal, please see our Privacy Policy .

Reviewer #1: **Yes: ** Brain C Chirombo, MBChB, MPH

---

## [Author Response · Author response to Decision Letter 1]

27 Jun 2025

Please, see Response Letter attached.

---

## [Editor Report · Decision Letter 1]

2 Jul 2025

Results of a fast-track HIV and hepatitis C screening protocol in Barcelona, Spain

PONE-D-24-57558R1

Dear Dr. Chivite Ferriz,

We’re pleased to inform you that your manuscript has been judged scientifically suitable for publication and will be formally accepted for publication once it meets all outstanding technical requirements.

Kind regards,

Yury E Khudyakov, PhD

Academic Editor

PLOS ONE
---

## [Editor Report · Acceptance letter]

PONE-D-24-57558R1

PLOS ONE

Dear Dr. Chivite Ferriz,

I'm pleased to inform you that your manuscript has been deemed suitable for publication in PLOS ONE. Congratulations! Your manuscript is now being handed over to our production team.

Kind regards,

on behalf of

Dr. Yury E Khudyakov

Academic Editor

PLOS ONE